# Optimization of an Industrial Sector Regulated by an International Treaty. The Case for Transportation of Perishable Foodstuff

**DOI:** 10.3390/e23010109

**Published:** 2021-01-15

**Authors:** Juan P. Martínez-Val, Alberto Ramos

**Affiliations:** 1Túnel de Frío, Fundación para el Fomento de la Innovación Industrial, 28006 Madrid, Spain; 2Department of Energy and Fuels, Universidad Politécnica de Madrid, 28003 Madrid, Spain; alberto.ramos@upm.es

**Keywords:** optimization, transportation, controlled temperature, isothermal tests, ATP

## Abstract

Transportation of perishable foodstuff is an engineering and commercial activity ruled by an international Agreement (the ATP) that needs an updated regulation. Before addressing such updating, some analyses are required about the physics of the problem, in order to identify the optimum use of the available technologies and the advantages represented by new methodologies that could be enabled soon. It is worth pointing out that manufacturers of ATP equipment follow quite closely the prescriptions given by this Agreement. So, optimizing those prescriptions will generate a general optimization trend in this sector. In this paper, a coherent analysis on these subjects is presented, and a new coefficient is proposed for qualifying ATP units, and some new tests are also proposed for measuring that coefficient in an efficient and inexpensive way. These goals are justified in this paper as a contribution from basic physics to a particular domain of Thermal Engineering. The paper is intended to be a bridge from Science to Technology, which is a must to get optimum results in exploiting technical knowledge.

## 1. Introduction

Optimization is usually featured as a mathematical process to find the minimum (or the maximum) of a function inside a given domain, either including or not additional constraints. It is worth pointing out that most of the effort on optimization has traditionally been put on the mathematical procedure to identify the point, in the phase space, where such a minimum value is attained. Even specialized books as Stoecker’s classical masterpiece, titled “Design of thermal systems” [1], is closer to a book on mathematics than a book on thermal energy. It happens more or less the same with Jalurias’ more modern book “Design and optimization of thermal systems” [2]. Both books are excellent by all accounts, but the core of optimization is the mathematical liturgy, which indeed is very useful for academic problems (usually defined as a set of equations with perfectly known coefficients).

Unfortunately, many real problems cannot be expressed as academic ones. This is particularly true for an industrial sector that moves huge amounts of money as it moves tons of goods. Only in Spain, around 50,000 certificates are issued every year for qualifying equipment for transporting perishable foodstuff. Close to 300,000 vehicles (from vans to lorries) are enlisted in this activity, which is ruled by an international treaty, as will be commented below. It is the ATP Treaty (short name for ” Agreement on the International Carriage of Perishable Foodstuffs and on the Special Equipment to Be Used for Such Carriage” [3]). However, the practical importance of this industrial and commercial field does not convey any requirement to optimize both the overall sector (the macroscopic scale) and the features of the individual equipment (microscopic level).

There is an urgent need to develop techniques to study how to improve the performance of real complex systems [4,5], as the one formerly cited. It actually presents some technical difficulties at the micro level, which can be treated with standard procedures of engineering optimization [6]. At the upper level, where the full integrated sector has to be taken into account, it still is more demanding, because of the difficulties to define the optimization objectives. It will be seen that qualitative logic can help in that quest, although neither the optimization at unit level nor the optimization as a regulated sector have so far been embodied in the international treaty. This is to some extent disappointing, because it seems that Thermal engineers devote a lot of time to learning how to solve academic problems, but they do not face practical ones.

It must be said that the optimization result cannot be better than its physical model [7,8], which in turn cannot be better than the accuracy of the coefficients describing the reality [9,10].

If those coefficients are not accurate, results are obviously not valid, which is something difficult to identify if there is not a previous indication of the order of magnitude of each variable. Besides the numerical value, another important feature of the relevant variables lies in their classification among boundary or external variables, independent ones and dependent ones, which is related to their physical nature. A variable that can easily be kept constant, as the thermal power given by an electric heater, can be classified as independent. On the contrary, temperature in any point must be classified as dependent, because the value at any moment is the result of the preceding thermal interactions, and it is much simpler to see it as an output from the system, while the power (either for heating or cooling) can be considered as an input [10].

When an engineering professional addresses the optimization of a system, he or she will start from a number of decisions already done (by the client, for instance, or by the ATP international treaty) but most of the problem will still stay undefined. Designing a system is to make the remaining decisions for the system to be unambiguously specified [11]. Those decisions usually will be limited by standards or, in our case, by a treaty. One of the aims of this paper is to analyze how far the cited ATP treaty is from the ideal scenario that could be considered the optimum one.

Of course, an overall optimization problem can involve different levels of analysis, and different domains to look for the solution, including feedbacks and conjugate calculations, but this can prove to be extremely time consuming [12,13]. Moreover, many engineering tasks are not specified as the search of a minimum (or a maximum) of a function but as defining a set of parameters that meet all the requirements of safety and quality [14] (in this context, quality means that the system complies with its intended functionality).

This is the case for classifying the equipment for transportation under controlled temperature. The subject has a lot of importance [15,16,17,18,19,20] because of the huge amount of perishable foodstuff transported every year under the auspices of the international treaty covering this activity in a large fraction of the planet, but the treaty does not embody any methodology with sound and suitable foundations for optimizing it.

Heat transfer is the dominant subject to that purpose [21,22], and the first thing to recognize is the high scientific and technical value of the available bibliography, with landmarking textbooks as the ones by Chapman [23], Incropera [24] and Lienhard [25]. Besides these ones, there also are a number of books and a larger number of articles dealing with the key problem for engineering activities: Physical Modelling [6]. Unfortunately, it will be seen in the following section that the ATP Treaty does not embody any sound and comprehensive physical model to establish the quantitative parts of the Treaty. So, Section 2 will be devoted to a short briefing on that Treaty, which will be our example for explaining how to use optimization in such technical services [26,27].

Section 3 will deal with the well-known basic equations of heat transfer, their boundary and initial conditions, and their topology, both in transients and steady-state cases. The objective of that section will be to connect the features required in the ATP equipment to the performance of the sector as a commercial supplier of specialized transportation services [28,29].

Those services require technology and this is the key element from the point of view of engineering. Optimizing the sector performance is equivalent to choosing the right technology, for the thermal transients of the load to be properly limited [30]. This is the subject of Section 4.

A technical discussion on the rationale to select optimum equipment for the new ATP test will be presented in Section 5. That discussion will embody elements for optimizing their performance.

The foregoing technical sections will be integrated in Section 6 to consider how to propose a deep change in the ATP treaty, including its qualifying tests [31,32,33]. This is the most important practical contribution of the paper that will have to be considered in the Treaty plenary meetings [3].

Section 7 presents the foundations of optimization in practical engineering, where equations have to merge with perceptions from stakeholders who are not familiar with Heat Transfer but must subscribe the best possible Treaty. This goal implies considering the problem in full scope. A main duty in engineering is to clearly state what the customer buys, and this is not addressed in the current ATP. In the optimized proposal presented in this paper, all these elements are integrated and expressed by a coefficient that answers the question of qualifying the ATP equipment in a meaningful way. This section can be considered as the main theoretical contribution of the paper, and it is a particular way to use information for optimizing engineering performance.

The concept of thermal coherence [34] will be used to this goal. A main prescription in this quest will be provided by the classical optimization rule of allotting the same effort to each degree of freedom involved in the definition of the optimum state.

The last section will present a summary of conclusions and a proposal of future work into two directions: first, to propose changes in the ATP Treaty; second, to widen the field of optimization in thermal engineering.

## 2. Briefing on the ATP Treaty

Under the auspices of the United Nations Organization, the treaty on International Carriage of Perishable Foodstuffs and on the Special Equipment to Be Used for Such Carriage (usually known as ATP treaty) was agreed and signed by first time in Geneva, Switzerland, in 1970 [3]. The General Secretariat of the Treaty officially is the UN Secretary General, although the administration and day-to-day secretariat was conferred to the United Nations Economic Committee for Europe (UNECE), particularly its Working Party 11. It must be noted that the treaty includes countries from other continents and is becoming larger [26].

When the Treaty was signed, interest in improving carriage conditions of sensitive merchandise was an important move in industrialized countries, because Temperature Conditioning Engineering had undergone an important advancement and emerging technologies at that time that created new expectations. In particular, Transfrigoroute [35] was an association of companies and professionals working in that field fifteen years before the birth of the treaty and was already a positive lobby for spreading information on these subjects, not easy to follow for many managers of standard moving fleets. Although Transfrigoroute was already at that time an important technical reference in that field, it seems to be that its influence in the technical parts of the Treaty was rather modest, although it still is an active and competent body, which is also committed to foster commercial competition even among its members.

The Treaty mainly was a political agreement, and it was conceived for including all type of possibilities in the different markets of the involved countries. Such a political root includes a condition in the Treaty that makes it very difficult to change a single word in it, namely, all variations must be approved unanimously. In general, this is close to impossible to reach in a meeting of the governing body of the treaty, because the specific interest of one country can be just the opposite of the proposing country. Nevertheless, some modifications have been approved recently, and they are already included in the current official version of the Treaty, valid since 6 July 2020 [3]. Unfortunately, most of the technical modifications of this new version do not follow at all any methodology for updating and optimizing the contents of the treaty. A critical review of its new version is actually needed, but this is outside the scope of the paper, and it requires a paper per se.

It is worth referring to the International Institute of Refrigeration (IIR) as the main independent intergovernmental science and technology based organization [36] that promotes knowledge in all refrigeration fields, including those related to ATP matters.

Transportation of perishable foodstuff is an important economic sector, which also affects public health directly. This means that authorities from different branches must participate in its management, although industry authorities have a particular domain to care of, namely the thermal performance of containers approved for this carriage.

The Treaty is ruled through the general meetings of UNECE WP 11 [26]. Although the history of ATP runs smoothly, its technical evolution is very weak, and its technical criteria have not been changed since the first wording. In the WP11 meeting in Geneva there have been several proposals to discuss about the origins of the limits of the global heat transfer coefficient of an ATP container (*k* limits). This coefficient must not be higher than 0.7 W/m^2^·K for the container to be accepted as ATP standard isothermal container and not higher than 0.4 W/m^2^·K for reinforced isothermal.

The actual problem is still more basic: does this coefficient represent the most relevant magnitude to qualify the thermal performance of an ATP container?

Some papers have been published on this subject with some contributions on different methods to calculate the *k* value [37,38] and about improvements on insulating panels [39,40,41,42,43,44]. These improvements should take into account the features of the service, particularly in the most complex space, which is the urban distribution [43].

In the evolution of this technology, some panels have included phase-change materials (PCM) which is a technique [45,46,47,48,49,50] closely connected with one of the categories of ATP vehicles, which use refrigerants inside the container as a heat sink.

Other containers use mechanical compression cryogenic cycles, so that the heat sink is not limited to a given mass of refrigerants. A complete solution can be found in a compound embodiment with thermodynamic cryogenics and PCM panels. There is already important experience on commercial fixed fridges [51,52] that could be transferred to transportation containers.

In any case, it seems that ATP will crash soon with the new regulations on refrigerants [53], and ATP will have to be fully rewritten, because it seems a rusted text as compared with new elaborations towards Sustainable Development. Very likely, the fall of current ATP will be produced by environmental requirements [54,55,56]. It has no sense to have a niche of the cold chain, the ATP, that does not follow the general rules. Once everyone accepts that the treaty must be fully changed, optimisation of the whole niche and detailed optimisation of different parts will be unavoidable to meet that objective.

Modelling the food distribution system has also been investigated [57,58,59], generally without taking into account the ATP Treaty. Moreover, new considerations are included in new scenarios [55,56], for taking into account energy consumption and environmental impact from frozen or chilled food transportation. Another important topic is the relation between the *k* value of a container and the actual result of an ATP carriage [56]. Some scenarios were considered for making a real distribution of perishable foodstuff, and the main conclusion was that other variables and boundary conditions had a much stronger effect on the temperature of each package at distribution. It is true that actual trips suffer from stochastic perturbations, which can produce significant delays. Moreover, seasonal and climate effect can also have a strong influence on thermal performance of the container, which can work properly in winter and deliver very bad service in summer.

The quest for thermal protection of food products has been addressed from different viewpoints, including classical approaches [60] and more complex systems including defrosting in combination with PCM [61] that can be used in installations for preserving perishable foodstuff.

The problem for an international Treaty is the need to keep some operational simplicity for not inducing strong differences among the parties. However, differences exist from some regions to others, and the selection of just one coefficient to qualify the container is a severe limitation for defining a good qualification method.

The ATP technical procedure is so simple that it does not convey any thermophysical analysis of the phenomenon being characterised. It just establishes some requirements for measuring the parameters involved in the definition of *k*, which are:A characteristic surface, *S*, corresponding to the geometric mean between the inner and outer surfaces. Measurements of those surfaces can suffer from ambiguity in some cases (wagons, tankers and enclosures with complex geometries). However, the current ATP Treaty considers that the magnitudes to be measured are clearly stated and do not have inherent physical problems. Surfaces are a characteristic of the unit, and they are not part of the experimental task. Those values are given by the builder. On the contrary, a main principle of a test procedure (according to the international standard EN17025, reference) is not to relay on data from other sources, unless already accepted by certified procedures. In other words, an ATP test station should make the geometric measurement of the internal and external surfaces of the body., but a new difficulty is found here in relation to both the Treaty and the cited standard: the ATP text ignores absolutely the standard (although most of parties oblige to follow the standard, as a routine of the corresponding Quality System of the country). In any case, a test stations must be responsible for the *k* value of every test and must measure all variables involved in the definition of *k*.Heating power (*Q*), particularly electric heating of the inside of the unit, which is the most frequent method in these tests. It must be noted that the ATP procedure can be breached in relation to this magnitude, because of the following gap: In annex 1 appendix 2, it is stated that electric energy consumption has to be measured with an error margin lower than 0.5%. However, there is not any explicit requirement to use this measurement to assess the accuracy of the recorded values of electric power, which will be used for the calculation of *k*.Difference in temperature (Δ*T*) between inner air (of the container or tank) and the outer air, which is the main variable in this subject (as happens in general in heat transfer processes) and must be measured in a steady state regime, with constant *Q*. Current ATP method requires the temperatures inside and outside the unit to remain constant for more than 12 h, which is much longer than the time span of the heat transfer transients across thermally insulated walls. As outer conditions must be kept constant for such a long time, the test to determine *k* must be carried out inside a test hall, fully equipped for keeping a constant temperature. Besides that, there must be an air flow alongside the container with a speed between 1 and 2 m/s. Inside the unit, some fans have to produce a total flow within the range of 40 through 60 times the internal volume of the unit, per hour.

The *k* coefficient is therefore calculated as:$$k=\frac{Q}{\Delta TS}$$

Both the internal and external speeds have a non-negligible impact on the *k* value, but they are not considered by any means in the *k* equation. This is somehow contradictory, because the coefficient *k* is not just the inverse of the thermal resistance of the conduction through the wall, but the inverse of that resistance plus those of the convection films on the internal and external surfaces of the unit wall. As this is a prescription of the test, it is not accounted for in the uncertainty analysis, but it is obvious that the thermal resistance of the convection films increases as the air speed decreases. This fact has an influence in qualifying a unit, and tests should take it into account. Nevertheless, an air-speed correction to the *k* value is not included in the test prescribed in the Treaty.

Besides the determination of *k*, the ATP includes additional tests to qualify the performance of cooling systems acting in the unit, as refrigerants (usually, eutectic material) and active refrigerating equipment, but they keep the *k* value as main reference.

It goes without saying that passive refrigerants and active refrigerators work in different time-dependent frameworks. For instance, a certain amount of refrigerant can absorb a certain heat load. Once such a threshold is trespassed, the refrigerant does not add any cooling help. On the contrary, an active refrigerator can work continuously, provided it has a source of energy. However, the time-dependent features of the unit are not used for characterising the unit.

In summary, the most severe criticism that can be said on the ATP Treaty, is that it does not provide to the users of those transportation means with an actual guarantee, or description at least, of the actual capability for keeping the merchandise under given thermal conditions. Nobody really knows what is the effect of using an ATP qualified equipment for a given purpose. This Treaty should be improved quite a lot by considering the full scope of the problem under study, which is how to keep a controlled temperature inside a transportation body.

### How to Optimise the ATP World?

The main objective of this paper is to try to contribute to a commercial sector that is growing at a high speed but in a spiral track. This industry is very much stagnated because the regulations of the governing Treaty are not appropriate for updating it. In fact, from the point of view of technical principles and engineering methodologies, the Treaty seems to be stopped somewhere in the past, and nobody seems to have the key to open the future. From the experience accumulated in the most veteran Test Station of Spain, it can be said that the Treaty is lagging behind important advancements in technology, particularly in the field of cryogenic cycles and the fluids used in them. Moreover, those thermodynamic working fluids, usually featured as R (refrigerants) have undergone a severe analysis in order to ban all of them having a large GWP (Global Warming Potential) or a large Ozone Depleting Potential (ODP) or both [53].

It is well known that many countries, particularly the USA through its Environmental Protection Agency (EPA) and the European Union, through the so-called “F-gas directive” [53] have approved very tough legislation on these matters. For a couple of decades there has been an increasing move towards more restrictions in the leakage annual rate that can be allowed, and the highest point so far in this trend has been achieved in 2020, because we have reached the largest list of banned refrigerants. Additionally, the amount of permitted gas used in an application has been limited to very low values.

Are those restrictions valid for all applications? No. As already said, ATP systems have not yet been included in the list, because the parties of the Treaty do not find how to move the ATP in the same direction as the legislation of Sustainable Development. It is a paradoxical situation, but it is what it is.

An optimisation of the ATP would have to follow the following stages, or similar ones:Clarification of the objectives of ATP.Compilation of available technologies useful to keep cold the inside of transportation units.Selection of the most suitable technologies to meet the established goals.Elaboration of a set of technical tests to verify the classification of equipment.

These tasks seem to be very different from the ordinary activities of mathematical optimisation, although they are absolutely necessary for establishing an adequate ATP framework.

About objectives, they are absolutely necessary for guiding the optimisation procedure at lower levels in the ATP structure. In the current version, the Treaty is just intended to give a few values on some variables in order to classify the transportation units, but the classification does not convey anything, is just a bureaucratic item. Saying that a truck has a coefficient K = 0.38 and it is therefore IR (insulated reinforced) does not inform the customer about the actual conditions inside the unit in a trip.

If an actually useful ATP, one of the first topics to be discussed is the identification of the coefficient to be used for classifying the equipment. Before addressing this critical point, some additional analysis on the physics of the problem are needed.

It must be explained that our procedure will be similar to a block chain diagram, in the sense that we will follow a string of blocks where a decision will be chosen and justified. From the point of view of technology, a compilation must be done, in order to find the optimum solution. There are three branches from that tree:Insulated unitsInsulated + passive coolingInsulated + active cooling

Although first-order differential equations will be used afterwards to study the thermal transients inside a transportation unit, it can be anticipated that “active cooling” is the only option that can guarantee to keep the temperature of the load in a given temperature range.

It goes without saying that, in some cases, heating is also needed (to avoid freezing of some foodstuff) but fortunately the existence of an internal combustion engine to power the truck is a guarantee of heating capability. In the periods of engine shutoff, either electric power or a small combustion heater can be used to that purpose.

So, the first decision in the optimisation process seems to lead to insulation plus active cooling (plus heating, but this is another chapter) and it must be underlined that the maturity of this technology has evolved quite impressively, from simple air conditioning units to large freezers. Reliability of cryogenic machines ranks among the first ones in ordinary machinery. Cryogenic gas leakage is a very rare accident.

Following the steps of this optimisation procedure, an important drawback can appear in the future, in relation to banned refrigerants. If the international agreements and regulations on this subject are taken into account, there seems to be a fundamental restriction to be met: a limited amount of refrigerant per application. This point could convey an important requirement for the ATP units, because they would have to work with limited refrigerant inventory. This problem can be addressed into two ways:Reinforcing the insulation to minimise the heat load coming in from outside.Designing and operating the cryogenic cycle so that the cooling power is maximised.

Both problems will be treated afterwards, once the relevant equations have been analysed. This analysis will also be used for specifying the tests for classifying the ATP equipment, which has to be a fundamental piece of the ATP puzzle.

The word transportation immediately implies thermal transients; and transients are characterised by parameters that cannot be measured in steady state conditions. A way of keeping constant temperatures inside the body or container is by applying active refrigeration, either by a compression cycle or an absorption cycle (although this one cannot reach temperatures as low as those from the former). These active methods can be complemented by temperature recording of the merchandise, either foodstuffs or pharmaceuticals. As a first proposal in relation to ATP, it can be said that ATP tests need complementary analysis, which must be found in the study of the heat transfer governing equations.

Additionally, the way a test is conducted, should be ruled by the already cited principle of thermal coherence, namely, the effort devoted to each parameter involved in the determination of *k*, should be proportional to the degree of accuracy attained in each type of measurement. (Usually, instead of prescribing accuracy, the test is defined with some limits in errors or uncertainty, but typology of errors must be carefully studied, because they are sources of additional problems in relation to the most critical part in the application of a Treaty or a standard, which is the acceptance criterion).

In the ATP, as in most of the standards, the acceptance criterion is twofold, of BEPU type (best estimate plus uncertainties): first, the best estimate of *k* has to comply with the threshold value characterising the category; and second, the error bar of the complete test procedure must be lower than +/−5% (in the last version approved of the Treaty, enforced on 6 July 2020, the wording was changed to say an expanded uncertainty lower than 5% with a confidence level larger than 95%, which is not fully equivalent. This change introduces a jump in the historical series of tests, but it is not a major problem, just a numerical adjustment).

If subindex 0 stands for the best estimate of the measurement of a variable, and δ stands for the error bar half-length, we can write, for instance:$$S=S_{0}\left(1\pm \delta_{s}\right)$$
$$k=\frac{W_{0}}{S_{0}\Delta T_{0}}\left(1\pm \delta_{k}\right)=k_{0}\left(1\pm \sqrt{\delta_{W}^{2}+\delta_{S}^{2}+\delta_{T}^{2}}\right)$$

The last equation shows the elements of the double criterion. $k_{0}$ is the best estimate for the coefficient, presuming the best estimates of the three measured variables have been corrected from biased errors, and the error bar half-length $\delta_{k}$ is a function of the error bars of the measurements. As all variables are involved in the *k* equation with the same weight or importance, the first optimisation rule is to declare that they have to have the same value, so:$$\delta_{W}=\delta_{S}=\delta_{T}<0.05/\sqrt{3}$$

This approach could be admitted, but it does include any reference to the effort needed to make a measurement, which has two components: the cost of the instrument and the time devoted to make the measurements, including the preparation phase, which in turn includes the calibration effort.

The ATP Treaty does not say anything about that, but this is not a specific problem. It is a general problem, and the guidance to properly solve it, is scarce. Moreover, it has to be connected with the type of variables under consideration, which can be classified into two categories:Independent variables, or inputs into the system, where we can classify both the Surface and the electric heating power, *Q*; they can be fixed along the test (or they are fixed by construction).Dependent variables, or outputs from the system, where we can classify the temperature map.

Temperature can be measured by different types of instruments, as thermocouples and thermistors, with a cost which is a very small fraction of the investment needed to build a test hall. This is an important cost reference. Another important reference is supplied by the time needed for the test, because it gives an indication of the price of the test. The ATP Treaty requires more than 12 h in steady state and a previous time of 6 h of preparation. So, ATP tests for determining *k* requires at least one day, which conveys a sizeable loss of money for the owner of the body. From an economic point of view, instruments do not seem to represent an economic burden.

The same can be said about watt-meters, which are forced by the Treaty to have an error lower than 0.5%. Additionally, temperatures should be known with an accuracy level better than 0.1 °C. This is equivalent to 0.4% error in a test where ΔT must be around 25 °C. Both measurements together imply an error of 0.65%, which means that measuring S can absorb an error of 0.49% without trespassing the overall limit of uncertainty.

Apparently, measuring distances is something that can be done with a high accuracy, but this is not so simple. For instance, measuring the diameter of a round table requires one to know where the centre of the circle is located. Otherwise, measurements will give a shorter distance, always, than the exact diameter. On the contrary. The distance between two parallel walls will always be measured longer than its actual value, because any deviation from perpendicularity will produce longer lines.

In the current version of the ATP Treaty, as in the former ones, treatment of geometric properties to improve the surface measurement is rather simplistic. It needs much better work. It is paradoxical that the major source of uncertainty in applying the Treaty stems from simple geometry, which can give much better recipes to measure surfaces, particularly when the body is made basically with flat plates. It must be noted that corrugated surfaces and the like are not accounted for in measuring the surface. In that case, it is only the flat virtual plate what is measured. However, the Treaty does not embody any method to make those measurements, which can be made after the following recipe, proposed by us:Each portion of the surface, either internal or external, is delimited by a contour where vertices are relevant elements. Three consecutive vertices make a triangle where two of the sides are coincident with two sides of the contour, and the third one is a diagonal going from the first vertex to the third one.So, we have three distances (three sides of the triangle) which can be measured easily with very high accuracy. Those measurements can be called *a* and *b* for the contour sides, and *d* for the diagonal.The area of the triangle is then:
$$A=\frac{ab}{2}\sqrt{1-{\left(\frac{d^{2}-a^{2}-b^{2}}{2ab}\right)}^{2}}$$

All walls of the body should be treated that way, and the final surface value would be the addition of all triangles covering the full surface without overlapping among them.

For curvaceous bodies, as tanks and vans, adequate geometric properties are more case dependent. There are some parts of bodies with circular sections, and many tanks are designed with elliptical cross sections. Although the recipe is not as simple as in the planar case, it can be stated that error in measuring surfaces can be made of the order of 0.5%, of the same order as measuring active electric power and temperature. A different topic is the time needed to make a test. It is too long, which is something not justified by any physical requirement.

Time is then the last relevant variable to be analysed. Before doing so, it is worth remembering (and putting together) some features of Heat Transfer transients.

## 3. Properties of Heat Transfer Equations

Optimising the use of knowledge to have a sound and coherent classification in a given engineering service, is a possibility that can save a lot of money, if properly done. Similarly, optimisation can be used to get the most in the information process aimed at defining qualifying standards for engineering equipment. In order to apply these principles to the field of transportation under regulated temperature, heat transfer equations must be analysed in their topological properties (within a general framework, for not to focus the analysis on a very detailed design of a very particular device).

Any textbook on Heat Transfer deals with heat conduction transients in a classical way, embodying the concept of thermal diffusivity, α, defined by the following equation, in terms of thermal conductivity, θ, density ρ, and specific heat, *C*,
$$\alpha =\frac{\theta }{\rho C}$$

It is worth recalling that ATP isothermal test lays on a steady state measurement, and the thermal diffusivity, α, does not play any role in that case. The steady state temperature profile within a solid is only expressed in terms of thermal conductivity, θ. We will restrict ourselves to the planar case, which has analytical solutions that can be used to guide the time dependent studies needed to characterise the physics of the problem under study. In the case of having several layers of thickness *H_i_*, an overall thermal resistance by conduction, *R_c_* is defined by:$$R_{c}=\sum_{i=1}\left(\frac{H_{i}}{\theta_{i}}\right)=\frac{H}{\theta }$$

The ATP qualifying coefficient for isothermal tests is the inverse of the total thermal resistance, from internal air to external air, *k*, which corresponds to the following,
$$k^{-1}=R=\frac{S}{h_{i}S_{i}}+R_{c}+\frac{S}{h_{e}S_{e}}$$

It is easy to identify that *S* stands for surface, *h* for convection film coefficient, and sub-index *i* and *e* for the internal part and the external part of the problem. In any convection film between a fluid and a solid planar wall, the following parameters are used:$T_{f}$ = temperature of the bulk of the fluid$T_{p}$ = temperature of the wall surface*h* = film coefficient*x* = space variable across the wall, (normal to the surface)$T_{\left(x,t\right)}$ = space-time dependent temperature within the wall. *x* = 0 is the origin and it corresponds to the surface; thus $T_{p}$ = $T_{0}$.

We will consider that the thermal flux, *q*″, comes from the left into the wall, and the boundary condition is then:$$q{}^{\prime \prime }=h\left(T_{f}-T_{p}\right)=-\theta {\left(\frac{\delta T}{\delta x}\right)}_{0}$$

Within the wall the time dependent conduction equation is:$$\frac{1}{\alpha }\frac{\delta T}{\delta t}=\frac{\delta^{2}T}{\delta x^{2}}$$

The structure of last equation implies that the product αt will appear always together. This product has dimensions of a surface and it can deassociated to a length, *l*, defined by:$$l^{2}=\alpha t$$

Similarly, from the boundary condition, Biot number, *Bi*, and Fourier number, *Fo*, are defined, for a wall with *H* thickness:$$Bi=\frac{hH}{\theta }$$
$$Fo=\frac{\alpha t}{H^{2}}=\frac{l^{2}}{H^{2}}$$

There are two classical cases that are used for the analysis of a thermal signal travelling inside a solid. Both of them relate to two infinite media, separated by a planar boundary. The half-space in the right-hand-side is filled with a solid, homogeneous material, with an uniform temperature $T_{u}$, at *t* = 0 and the left hand side is defined by the boundary condition. In the first case, the half-space is filled with a fluid at temperature $T_{f}$ and it interacts with the solid through a film coefficient *h*.

In the second case, the half-space is empty, and a thermal radiation flux *q*${}_{0}^{\prime \prime }$ impinges normally onto the solid surface.

Solutions are given in the following equations, were “erfc” is the complementary error function (which is included in Excel sheets and many more mathematical compilations).

Temperature evolution in the first and second cases are given by:$$\frac{T\left(x,l\right)-T_{u}}{T_{f}-T_{u}}=\mathrm{erfc}\frac{x}{2l}-\left(\exp\left(Bi\frac{x}{H}+Bi^{2}Fo\right)\cdot \mathrm{erfc}\left(\frac{x}{2l}+\frac{hl}{\theta }\right)\right)$$
$$\frac{T\left(x,l\right)-T_{u}}{\Delta T_{0}}=\frac{2}{\sqrt{\pi }}\sqrt{Fo}\cdot \exp\left(-\frac{x^{2}}{4H^{2}}\frac{1}{Fo}\right)-\frac{x}{H}\cdot \mathrm{erfc}\left(\frac{x}{2H\sqrt{Fo}}\right)$$

*Bi* and *Fo* have been used with thickness *H* as characteristic length, because in our case, there is not such a solid material beyond that point. There we find a second boundary, where a colder fluid (the external air) will remove the arriving heat, although the cold boundary condition has to be coherent with the film coefficient and the temperature of the fluid. This means that there will be a heating wave front moving eastward, and a cooling wave front moving westward. Both of them will adjust to a continuous straight line inside the solid wall.

Table 1 presents the results of last equation for *x* = *H* (the opposite face of the slab, which is 0.1 m thick and has a thermal diffusivity of 0.57 mm^2^/s). The arrival time of the thermal signal is found around *l* = **0.035 m**; which means 2150 s in this case, with an insulator of very low diffusivity. Higher diffusivities would lead to shorter times. The transient in this case could be considered finished at *l* = **0.05 m** (4400 s), because the increase in temperature in the hot face will be around 20 times as high as the increase in the cold face, taken the chamber temperature as reference (this means 5% in the last by one column).

The last equation can also be formulated in terms of nondimensional variables, with the following definitions:$$\varphi =\frac{x}{H}$$
$$\tau =\frac{l}{H}=\frac{\sqrt{\alpha t}}{H}=\sqrt{Fo}$$
$$\frac{T\left(\varphi ,\tau \right)-T_{u}}{\Delta T_{0}}=\frac{2\tau }{\sqrt{\pi }}\cdot \exp\left(-\frac{\varphi^{2}}{4\tau^{2}}\right)-\varphi \cdot \mathrm{erfc}\left(\frac{\varphi }{2\tau }\right)$$

This equation represents an infinite set of self-similar solutions that are summarised in just one nondimensional solution. Of course, the projection of this solution onto the real world gives a particular expression for each case, governed by its real values of conductivity, density and specific heat.

In this equation, $\Delta T_{0}$ is the temperature difference that appears between the surfaces of the wall of thickness *H* and conductivity θ, when a thermal flux $q{{}^{\prime \prime }}_{0}$ goes through it, in steady state. This is governed by Fourier’s law,
$$q{{}^{\prime \prime }}_{0}=\theta \left(\frac{\Delta T_{0}}{H}\right)$$
ATP tests are based on the previous equation, plus the thermal resistance of the convection films, but the main contribution comes from the resistance of the conduction term, which is (*H*/θ).

The former nondimensional equation indicates that the heating wave front arrives at a depth H in the solid half-space when τ = 0.4. Therefore, Fourier number is 0.16. This value can change slightly from one author to another, because of small differences in defining the arrival time. For instance, Kaviani’s textbook gives *Fo* = 0.17 for the arrival time (which is not the end of the transient, but the beginning of the end).

Before searching for an optimised methodology using the most adequate variables to characterise the system, some additional features of heat transfer mechanisms must be embodied in our analysis, in order to have a complete view of this problem.

### Retardants and Insulators

Anyone can wonder about the very large difference in the time scale of the original ATP method and the alternative one. The former requires more than 18 h of constant figures in the recorders and only 12 readings per hour; and the latter needs measuring temperatures, power and film coefficients every minute, but only for 1 h, once the relevant variables become stabilised (which mainly depends on conditioning the test hall).

The difference is rooted in the lack of analysis supporting the ATP procedures. It can be presumed that some rationale was followed fifty years ago, in the first redaction of the Treaty, but it was not included as an annex.

Indeed, an important element was the type of material chosen to characterise the ATP basic problem. It seems that “retardants” were used to select the time scale of the test, while the ATP classification was based on low values of conductivity, and this is the case of other type of materials: “insulators”. A typical retardant is water, with a very low thermal diffusivity, of 0.15 mm^2^/s. Water conductivity is fairly large (for insulating purposes) 0.65 W/m·°C, and its “retardant” properties come from density and specific heat, which have very large values, 10^3^ kg/m^3^ and 4.2 × 10^3^ J/kg·°C.

An actual insulator (polyurethane, polystyrene, …) has a conductivity of 0.04 W/m·°C (16 times as low as the one of water) and a thermal diffusivity of 0.60 mm^2^/s (in round numbers), which is 4 times as high as the value of water.

This means that thermal transients in insulators are much shorter than in retardants. For instance, for a wall 10 cm thick, a Fourier number of 0.4 would be reached in 6600 s (less than 2 h) in an insulator, and it would be more than 7 h in water (a retardant). From this point of view, retardants seems more appropriate for avoiding heat transfer from inside to outside the body. That is true, but there is a hidden problem that can disqualify classical retardants, as water, as a wall material. Note that the ATP classification is stated in terms of the overall heat transfer coefficient, which is the inverse of the thermal resistance (which must be higher than 2.5 m^2^·°C/W for qualifying as IR). In insulators, a thickness of 0.1 m with a conductivity of 0.04 W/m·°C just gives a resistance of 2.5 (=0.1/0.04). On the contrary, water would give a resistance of 0.15 (negligible).

In summary, the original ATP was thought for insulators, but the time scale of retardants was retained in the definition, so asking for an exceedingly long time span for the test.

## 4. The Thermal Transient Inside the Container Chamber of an Atp Unit

The material that must be protected in an ATP trip is not the walls of the container, but its internal charge. So, we must look for some advice from the evolution undergone by the charge of the unit. It is obvious that the charge can have very different characteristics, but a familiar and common example will help us for describing this point.

We consider milk occupying the internal chamber of the container. We make that selection because of two reasons:Milk has a very high thermal inertia (similar to water) and could absorb easily all the heat arriving to the inner face of the insulating panel. This point is related to the balance of enthalpy.A fluid can have internal circulation either by natural convection or by forced movement (with a mixing propeller, for instance) what conveys the idea of fast uniformization of temperature inside it, and a powerful transfer mechanism in the boundary panel-milk. This point is related to the heat transfer balance.

Any thermal problem must fulfil both balances. We simply add that a fluid can move, and that is enough to support our assumption of fast internal uniformization in temperature. We must now examine the transient inside the chamber, which will depend on the category of the vehicle. This analysis will also help us point out the merits of each category.

The interface between the wall and the contents of the chamber can present different features that could be summarised as follows:Solid–liquid interface, which usually conveys a relatively high film coefficient.Solid–gas interface, where the film coefficient is smaller. This case is very common, because of the air filling the empty spaces inside the chamber.Solid–solid interface, which is not frequent, but should be considered.

If we start by last one, we note that, in the interface between those solids, there will be a discontinuity in diffusivity. If this one is larger in the charge than in the wall (which would be the common case) the thermal wave front will accelerate inwards, and the actual limitation in heat transmission would remain in the insulator. On the contrary, if the diffusivity of the internal load is smaller, the wave front will decelerate, and a second insulating mechanism will appear (the second insulator being the internal load). Nevertheless, a physical situation like this one is not fully credible, because both solid, panel and load, would have to be in very close contact, and this is not the common case. Air will be filling any separation between both solid surfaces.

The most credible case is therefore wall-air-load. Natural convection will exist, but we will disregard it for the moment. This means that we will have in the chamber a gap of stuck air between the panel and the load. Suppose in our numerical example that 5% of the volume of the chamber (of the order of 3 m^3^) is filled with air and that volume is in contact with the wall. As the surface of the interface is about 100 m^2^, we can consider that there is an intermediate “panel made of air” with a thickness of 3 cm.

The diffusivity of an insulator is around 10^−6^ m^2^/s. (This can be justified by an insulator with a conductivity 0.04 W/m·K; a density of 40 kg/m^3^; and a specific heat of 1000 J/kg·K). In the case of air we have a conductivity of 0.024 W/m·K; a density of 1.2 kg/m^3^; and a specific heat of 1000 J/kg·K; which means a diffusivity of 2 × 10^−5^ m^2^/s. This value is 20 times as big as the diffusivity of the panel. This means that in the worst case for heat transfer, without air movement, the heat transfer mechanism in air will have a rate high enough to absorb the heat arriving at the interface, but this is not enough. The balance of enthalpy points out that air has a very low thermal inertia, and the result will be that all the heat arriving to the panel-air internal interface will be transferred to the (solid) load.

Solid material inside the container can be placed in different frames, from small independent pieces to a big single hyperblock, but they share a common element: air filling all free spaces inside the container. It is the heat carrier fluid between the hot parts (the inner face of the insulating wall) and the outer coatings or the skins of the load. It evolves according to a law that can be expressed as follows, in a lumped-parameter coarse approach that does not consider the time-space dependence of the actual problem, but gives very clear indications of the physics of the problem,
$$\rho VC\frac{dT}{dt}=S_{p}h_{p}\left(T_{p}-T\right)-S_{c}h_{c}\left(T-T_{c}\right)$$
where ρ, *V*, *C* and *T* refer to air inside the container, subscript p refer to the panel (or wall) and c refers to the internal load.

The delay induced by the air (of the order of 10 s) will be very short in comparison to conduction mechanisms in solids (panel and load, with delays over 100 s). Besides that, the heat capacity of inner air (per °C) is much smaller than the value we find for milk, where:$$\rho VC=940\times 60\times 3.9=220\;\mathrm{MJ}/{}^{\circ }C$$
And for air:$$\rho VC=1.2\times 3\times 1=3.6\;\mathrm{kJ}/{}^{\circ }C$$
Therefore, the temperature of the air will stabilise soon, and the heat transfer balance could be written as a function of temperatures in the panel inner face ($T_{p}$), in the bulk of the chamber air ($T_{a}$), and in the load surface ($T_{c}$).
$$S_{p}h_{p}\left(T_{p}-T_{a}\right)=S_{c}h_{c}\left(T_{a}-T_{c}\right)$$
The former equation seems very simple, but it embodies the enormous difficulties to calculate properly the film coefficients, which can vary quite a lot from one place to another. Nevertheless, it conveys the idea of having an internal “heat buffer” (air) which redistributes the incoming heat from the hotter surfaces (the panels) to the cooler parts (the load) of the trajectories or circuits air undergoes inside the container chamber.

About the case of solid–liquid interface, the example of milk is still valid, although the container would have in this case a different form, and it will be a tank instead of a box.

In each piece (dS) of the inner surface of the panel, the heat transfer equation will be:$$q\left(x_{p}\right)=-\theta \frac{dT}{dx}=h_{p}\left(T_{p}-T\right)$$

And the total enthalpy balance will be:$$\rho VC\frac{dT}{dt}=S_{p}h_{p}\left(T_{p}-T\right)=S_{p}q{\left(x_{p}\right)}_{average}$$

*q*($x_{p}$) is the incoming heat from outside.

The foregoing equations must be applied to the three branches we identified in ATP:Plain insulation.Passive refrigeration (by Phase Change Materials)Active cooling (by a thermodynamic cycle)

The first category is mandatory for the others. A characteristic evolution of the temperature profile inside the panel is depicted if Figure 1 (for the insulator data given before, when diffusivity was introduced).

As the thermal flux at any point is a function of the T gradient at that point, Figure 1 suggest to compare the thermal flux coming into the panel (by the outer face) and the flux going out of the panel through its inner face, at a given moment. This information is given in Figure 2. It can be seen that *q*(0) is a straight line (in log-log representation). At the beginning, it is much larger than *q*(0.1 m) because of the differences in gradient, as already shown in Figure 1, for *t* = 1000 s, for instance. However, for 10,000 s (which is 0.01 in the αt scale) the slope of T inside the panel is almost constant and both q values are very similar. As seen in Figure 2, there is an asymptotic trend in *q*(*x* = 0.1) t merge with *q*(0).

When both *q* values become almost equal, the role of the insulator vanishes. The flux arriving into the container chamber is the same as the flux incoming by the outermost surface. So, an “Insulator Performance Indicator”, $I_{p}$, can be defined within this context as:$$I_{p}=1-\frac{q(0.1)}{q(0)}$$

## 5. The Rationale for an ATP with Optimum Carriages

It is seen if Figure 2 that this indicator is 1 at the beginning, and it decreases at a fast rate. It can also be used to define a criterion for the quality of the insulation. This point allows us to come back to the categories of the treaty.

### 5.1. Plain Insulation

Former figures are really relevant to this case and the introduction of Ip is a reasonable way to show the time-dependent character of a problem that is treated in ATP as a steady-state. A situation as the one created in the test stations to determine K values does not correspond to the actual case of plain insulation. Moreover, the worth of the insulator as such decreases with time, as shown in previous figures, and this means that a thicker insulator would be needed if a longer period is required for proper protection of the foodstuff.

It is worth reminding that *q*(0) is given by:$$q\left(0\right)=\frac{\theta (T_{0}-T_{i})}{\sqrt{\pi \alpha t}}$$

Note that it has a singularity for *t* = 0, because the boundary and initial conditions imply a jump in the outer face border, but its effect on the rest of the time and on the integral value *E* is negligible.

In fact, *E* (time integral of *q*) can be written as:$$E_{0}\left(t\right)=\frac{2}{\sqrt{\pi }}\rho C\left(T_{0}-T_{i}\right)\sqrt{\alpha t}$$

The problem with Plain Insulation is that the load will be heated by external conditions, if the atmosphere is hot and/or the time inside the container is too long. The temperature profile inside the container, including the effect of the air filling all gaps in the chamber, is shown in Figure 3.

The inevitable trend to isothermalization makes useless the insulation in the long term, as pointed out by the indicator $I_{p}$. A possible criterion for qualifying the containers would be based in limiting the distance and the time span of carriage operations depending on the Ip value; but this proposal lays beyond the scope of this work.

### 5.2. Passive Refrigeration (By Phase Change Materials)

As long as the PCM is partially frozen, those panels represent a point of constant temperature. Having a large enough equipment of PCM panels, with a proper level of phase change temperature, can help maintain the carried foodstuff under good conditions.

Note in Figure 4 that in this case the performance of the insulator can be much more explicitly defined, because the temperatures at both sides are fixed. As a matter of fact, the worst situation to be considered in this case is that the container walls (or panels) are at the beginning at the same temperature of the environment (say 25 °C) and therefore the internal PCM panel launches a cooling wave front outwards, in reverse deployment as the one studied in the Plain Insulation category. Moreover, after the first transient, the profile of the temperature will become very close to that of Figure 4, and the flux will be calculated by immediate application of Fourier’s law. It holds:$$q\left(x=0.1\right)=\frac{0.04}{0.1}\left(25-1\right)=9.6\;W/m^{2}$$

This value can be understood with the help of Figure 1, referred to plain insulation. It is seen that the evolution of the profile in that figure goes toward straight line of a slope of 10–12 °C/0.1 m. With a conductivity of 0.04 W/m·°C, the result is lower than 0.5 W/m^2^.

The problem is that a cooling wave front moving outwards was not included in the Plain Insulation case, because of the interest in just studying the insulation effect. Nevertheless, if that problem is complemented with a cold load inside, acting as a heat sink, the heat incoming flux of the Plain Insulation case would increase quite a lot.

Going back to the passive cooling of Figure 4, it is very important to point out that the “heat inverse losses” previously calculated (9.6 W/m^2^) are supported by the refrigerants (or PCM) not by the load. If we consider the PCM is ice, said losses would convey a rate of thaw of 0.03 g/m^2^·s, which is slightly over 100 g/m^2^·hour. In 10 h (around a labor day) it would be 1 kg/m^2^. This value corresponds to a layer 1 mm thick, which is very small. In fact, an operational ice panel will have a thickness of 2 cm at least.

There is abundant technical literature on latent heat applications, including those for trucks at low temperatures [39]. About hybrid isolation panels, with some mixing of PCM pieces inside of standard insulators, there are some proposals [42,43,44] that look for an apparent increase of the specific heat of the panel, what is a way or reducing the effective diffusivity. Therefore, the panels show a longer period of effective insulation. Experimental evidence on this technology is starting to appear [41]. However, phase change materials must be selected carefully, because they always behave as points of fixed temperature. This is not a complex problem, because of the many different PCM substances already identified as PCM to be used in this field [40]. Of course, one of the most appealing material that could be considered as the first PCM material in an ATP refrigerated container, is ice, that can be extended to aqueous salt solutions [45] which offer a large variety of materials that could fit in many applications.

### 5.3. Active Cooling (By Thermodynamic Cycle)

Heating, ventilation and air conditioning (HVAC) is a traditional field of work of Thermal Engineering, which has evolved quite a lot in recent decades and offers a broad set of solutions for almost every kind of application. In particular, this is the case of ATP vehicles, which are designed nowadays with very accurate tools [38].

From the point of view of the general assessment we are presenting in this paper, active Cooling is the superior category, because it is the only one that can give flexible and powerful answers.

Figure 5 shows the schematic profile of T in the external (or insulating shell) of an ATP container of this kind. It is to some extent similar to the Passive cooling by refrigerant panels, with the obvious differences on the heat sink and cooling mechanisms. The thermodynamic cycle has to provide a cooling effect of certain power at a given temperature. For instance, in the former example of Figure 4, repeated in Figure 5 with variations in the heat sink, there was a heat transfer rate of 10 W/m^2^·°C, which implies just 1 kW (for 100 m^2^) of cooling effect, that would take place in the evaporator of the compression cycle, that would work at −5 °C or somehow lower T. This is absolutely standard at present, and even evaporator temperatures around or below −20 °C can be considered as household values.

In summary, active cooling technologies present the highest degree of flexibility for meeting the objectives of ATP and are based on standard equipment that has already incorporated the advancements in Thermal Engineering.

Another interesting possibility is to have all three pieces of equipment: insulators (this is already mandatory), passive refrigeration (with PCM) and active cooling.

Insulation (with a low conductivity) is a must when there are fixed temperatures at both sides (inner and outer) of the container wall. PCM has the role of a “cold storage” that can be very useful to increase reliability and flexibility [46,49] of the container performance. Furthermore, active cooling provides the capability to react against any negative condition, usually an increase in external temperature over the expected level. In fact, such hybrid scheme is already working in air conditioning [49,50] and commercial fridges [51,52]. A detailed analysis complemented with experimental work [42] is very illustrative to this purpose.

However, the need of reloading the cold charge of the passive cooling panels make them not very attractive. In a qualitative review of the number of ATP units including PCM panels, it was clear that they are loosing ground in this niche.

This leads our optimisation analysis to select the following class as the optimum one for the future ATP: reinforced insulation with active cooling. (We can add low gas inventory machine for the cryogenic cycle).

## 6. Redefining the ATP Treaty and Its Technical Annexes

Redefining the ATP Treaty will require:To identify the requirements for the carriages to be used, which has been the subject of the preceding section.To identify coefficients representing quite accurately the quality of a unit for the functionality of ATP (this is the topic to be addressed here).

Before attempting that quest, it could be useful to elaborate something else on the type of mismatching found between Physics and Treaty. In particular, let us consider the following definition and criterion of acceptance (Annex 1, 1st clause):

“*Heavily insulated equipment specified by: a K coefficient equal to or less than 0.40 W/m^2^·K and by side-walls with a thickness of at least 45 mm for transport equipment of a width greater than 2.50 m.*”

The second part of the criterion conveys the idea that a given level of protection requires a proportionality between the size of the container and the thickness of the insulating panel, and this is not so. Just the contrary, when thermal inertia (a concept hardly considered in the annexes of the Treaty) is properly taken into account, the protection of the foodstuff, up to a given level, imposes thicker panels for smaller chambers, because the smaller the chamber, the smaller the ratio between thermal inertia and heating surface.

In order to keep the explanation in analytical form, we will keep the example of a tank filled with milk, and we will consider the tank to be cylindrical, with a circular cross section. For the sake of keeping similarity among the tanks, we include the following condition between length *L* and radius *R*.
L=10R

We are going to simplify the problem by considering that the thermal flux coming into the chamber follows the “thin wall” model; and we will also assume that the temperature profile inside the panel is linear. (In a cylindrical wall the T profile in steady state conditions is logarithmic. However, being a thin wall, the logarithmic expression can be expanded in a Taylor series, and it becomes linear).

We will also include simplifications disregarding the convection mechanisms in both faces of the panel, particularly in the inner face, where we can write:$$T_{i}=T$$
*T* being the average temperature of the milk, $T_{i}$ the temperature at the inner face of the panel. This equation is an oversimplification because it implies that the convection mechanism has an infinite film coefficient, but it does not perturb the transfer of heat. In the outer face, $T_{0}$ is kept constant.

Then we can write the following balance equation, where ρ is the milk density, *C* its specific heat, and *V* the total internal volume. The left-hand-side is the variation of milt temperature, and the r-h-s is the total incoming thermal flux:$$\rho VC\frac{dT}{dt}=\frac{\theta }{H}\left(T_{o}-T_{i}\right)S$$
where *H* is the thickness of the wall (or insulating panel) with a conductivity θ, and,
$$V=\pi R^{2}L=10\pi R^{3}$$
$$S=2\pi R^{2}+2\pi RL=22\pi R^{2}$$
The balance can thus be rewritten as:$$\frac{dT}{dt}=\frac{\theta }{H}\frac{22}{10R}\frac{1}{\rho C}\left(T_{o}-T_{i}\right)$$
The origin of temperature scale can be fixed at the constant value $T_{0}$ and we define this scale by γ:$$\gamma =T_{o}-T$$
$$\frac{d\gamma }{dt}=-\frac{dT}{dt}$$
$$T_{0}-T_{i}=T+\gamma -T=\gamma$$
Therefore, we can write:$$\frac{d\gamma }{dt}=-\frac{\theta }{H}\frac{22}{10R}\frac{1}{\rho C}\gamma =-\frac{1}{HR}\frac{22\theta }{10\rho C}\gamma$$

Variables corresponding to geometry are only *HR*, which go together. Note that the factor multiplying γ is the inverse of a time, that can be called ω. The solution of γ will be:$$\gamma \left(t\right)=\gamma_{0}\cdot e^{-\omega t}$$
where $\gamma_{0}$ is equal to ($T_{0}$-$T_{i}$) at *t* = 0. And:$$\omega =\frac{1}{HR}\frac{22\theta }{10\rho C}$$
with previous data taken for milk (940 kg/m${}^{3}$; and 3.92 kJ/kg·K) and θ = 0.04 W/m·K, for the conductivity of the panel, it holds:$$\omega =\frac{1}{HR}2.4\times 10^{-8}$$
For *H* = 0.1 m and *R* = 1 m, ω = 2.4 × 10${}^{-7}$ s${}^{-1}$. Note that the evolution is very slow because of the high thermal inertia of the milk and the low conductivity of the insulator.

What matters in this case is that *H* and *R* go as a factor, and therefore if the same time evolution is sought in a big tank and a small tank, the latter will have to have a thicker insulator thickness, *H*, than the big tank. This is not a paradox of thermal engineering, but a physical consequence of thermal laws. Namely, thermal inertia goes with *V* and the heat load goes with *S*. To some extent, the criterion on reinforced insulation is misleading. Even more, the criterion for normal insulation does not contain any reference to the thickness.

This discrepancy between Physics and Treaty is of uttermost importance for identifying the coefficient to characterise the new ATP units, that can be based partially on the old treaty, for not to make a revolution where technical evolution and clarification is more than enough. The new coefficient, *A*, would be:$$A=K\cdot S=\frac{Q}{\Delta T}$$
*A* is expressed in W/°C, and it means the cooling power that must applied to the inside of the unit to keep it with a temperature difference Δ*T* from outside. Of course, an immediate use of this coefficient is to calculate the required power of the active cooling to satisfy a given situation. It is important to realize that it is not necessary to determine neither *K* nor *S*.

So, *A* would be the appropriate coefficient to qualify a unit, but it is an absolute coefficient, that does not take into account the volume of the unit. It is worth repeating that *A* is what matters for qualifying a unit, and for integrate in it a cryogenic cycle of enough power. However, if we want to have a specific coefficient to make comparisons, we could used coefficient *B*, defined as:$$B=\frac{A}{V}=\frac{Q}{V\Delta T}$$

*B* gives the cooling power needed to keep a difference Δ*T* per cubic meter of container.

Another very important variable in ATP tests, is time. Current specifications of tests imply very long testing periods (longer than one day) which is utterly unnecessary in the new ATP definition proposed in this paper.

In order to support this idea, and to specify a new type of test, some new tests were carried out at Getafe ATP Test Station, according to the following procedure:Measure and register all the temperatures every minute, and then calculate the average for a time span of 5 min, so that ATP data and new data can be compared.Take the first interval of 1 h in the new data that shows stabilised data in all temperatures. “Stabilised data” means that heating power does not have variations higher than +/−1%; individual temperatures averaged every 5 min do not have variations higher than +/−2 °C; and the difference between average temperatures of the internal and external air do not have variations higher than +/−1 °C. Calculate the overall heat transfer coefficient *k* using the averaged values of the heating power inside the body and the temperature difference between the inside and the outside of the body. Compare this result with the one obtained from current ATP methodology, which requires at least 18 h but temperature is recorded only 12 times an hour.

In the new procedure proposed, 1 h is enough to reach the values to calculate *k*, (after stabilisation, which is a fundamental first stage of the test, and it is particularly depending on the hall, because of its large thermal inertia).

In order to guarantee that actual stabilisation has been reached, the new procedure requires one to measure and register the temperature on the central points of the inner and outer faces of 3 sides of the unit. These temperatures are not part of the determination of *k*, but they help ensure that the steady state has been reached in the unit walls. They must not have variations higher than +/−2 °C in the stabilised hour.

Additionally, the new procedure should include the measurement of the local value of the film coefficient in the proximity of the central points of the faces where temperature is measured. Those local values of the film coefficient do not participate directly in the calculation of *k*, but they are an additional warranty of the steady state dominating the full thermal system. Taking into account that the local value of the film coefficient can oscillate more (because of turbulence, for instance) than the temperature, variations permitted during the stabilised 1 h period can be +/−10%.

The rationale for the new proposed procedure rests on measuring along a time span and with a frequency much closer to the features of the thermal system, rather than the current ATP method, which requires too long times and does not convey any significant information for the users of ATP services. Note that the current method conveys a very poor knowledge of the evolution of the temperatures, which are recorded every 5 min.

The new procedure presented in this paper conveys a justified change in time and frequency: no longer than 1 min to register; no longer than needed for the test; which must include the measurements of more temperatures and some local values of a coefficient that govern quite a lot the heat transfer, namely, the convective film coefficient.

In the following Table 2, Table 3, Table 4, Table 5 and Table 6, results are given of a comparison between the results of standard ATP tests and results with the new specifications. Test were conducted in Getafe ATP Test Station. The VIN (Vehicle identification number) is given first, followed by a table with the heading, a first row that is the standard test, and a second row, corresponding to the shorter procedure, plus a third row, which is the ratio between the shorter method result, divided by the standard method result (expressed in %). They are practically the same, always. Labels in the upper row mean the following:Ti = inner air temperatureTe = outer air temperatureDTi = inner air, internal temperature differenceDTe = 0uter air, internal temperature differencePTV = fan powerPTR = electric heating powerPT = total heating powerDtie = temperature difference between inner and outer airTm = mean temperature inside the unit wall (mean temperature between inner and outer air)

### Result of Tests

In summary, there seems to be a sound technical justification for establishing two new coefficients A and B for qualifying ATP units, and an alternative shorter procedure for the determination of the overall heat transfer coefficient (in the current ATP methodology) or the new proposed coefficients, A and B, much better fitted to the real needs of this complex thermal service.

## 7. Optimising Engineering Prescriptions

Technology has evolved enormously in 50 years, and this is something utterly ignored in the ATP Treaty, which requires the acceptance of all the parties for approving any amendment. Nobody is talking on behalf of technology in the annual ATP meetings, but anyone can stop any trend towards modernisation for any reason (or without any reason at all). So, a new ATP seems to be needed, updating technology and including a way to give commercial and social value to the engineering prescriptions related to the new ATP. This requirement implies that any person dealing with a carriage having a “new ATP label” would know clearly what is the service this carriage can provide.

If both concepts, technology and commercial value are put together in an engineering scheme where optimisation tools will be used to get the most from them, the main tag epitomising a qualitative scenario for such an optimum is “temperature control”. This means 100% of the active time of the carriage not just a certain interval defined in a test.

This ambitious objective is not ambitious at all. Air conditioning units, chillers and freezers are today absolutely household names in many countries of the globe, and the rest of the countries (with lower technology development) also receive the benefits of such evolution, although at a slower speed.

A 100% temperature control needs above all a heat source and a heat sink (it does not mean that both elements have to work all the time. Usually only one will be needed at a time).

The heat source will be the engine of the carriage, if it is an Internal Combustion Engine, or the batteries, if it is an electric vehicle. Besides that, when the carriage has to stop for a long while, the heat source will come from the ordinary electric grid, which will have to provide this service in a growing pace, because of the increasing importance of electricity in transportation.

The heat sink is more specific and cannot be related to essential items for moving the carriage, but that sink will be provided by the evaporator of a cryogenic cycle working at the required temperature levels. Although other options could be considered for some specific cases, the general solution will be given by compression cycles. This technology is fully dominated by car makers and repairing workshops, and the level of confidence in this technology is very high. The compressor will be powered by the mechanical transmission of the carriage, if it is an ICE machine, or by an electric engine, if the carriage is powered by batteries or is connected to the grid.

All the elements are therefore available for making a quantum leap in ATP. Optimization in engineering design and operation requires to have degrees of freedom to decide, and this is the scenario we have just put in perspective in the foregoing paragraphs.

Updating and redefining the ATP Treaty will convey to pay attention to a series of important topics, particularly on environmental requirements [53,54,55,56]. The reality of transportation features [57,58,59,60,61] will also require specific considerations.

Two points still need some attention to close this proposal: economics (which is essential in optimisation) and formulation of the new ATP test. Both of them are connected.

About the optimisation scenario, things can be elaborated as follows:The base of the carriage is an engine on a chassis plus a cabin. This is the most expensive part of the total machine, by far and large. It can be associated to a capital cost of 100 au (arbitrary units, for instance, €) and a reduced lifetime (expressing the income along time in its corresponding present worth) of 20 years. This means a capex of 5 au/year. It is also necessary to account for fuel cost, maintenance and other operational costs, that can be around a similar amount, with a total expenditure of 10 au/year.The second element is the insulated body, which is much cheaper than the previous element, with a capex of 10 au, which has to be distributed along its lifetime, so yielding 0.5 au/year (maintenance is very simple and cheap, and there is not any other opex to be taken into account).The third element is the cooling machine. A capex of 20 au can be associated to this element, with a reduced lifetime of 20 years, and very small opex, because freezer and chiller maintenance are totally mastered by commercial technicians, and this cost can be considered already included in the guarantee.

So the total annual cost of the carriage would be around 11.5 au, which has to be compensated by the commercial services the carriage can do under such a new scenario for a new ATP acting after the proposed methodology presented above.

The main advantage is that the carriage for transporting perishable foodstuff and other products that need controlled temperature, would have a clear new ATP category, given by the result of its test (or the test of the prototype unit of that series). The new category could have a continuous-value indicator, or several, as the coefficient of performance (COP) of the cooling machine for some specific conditions, relevant for this type of commercial services. For instance, to keep 20 °C below zero inside the isolated and cooled body, while the external air (confined in the test hall) is at 40 °C.

The former example would correspond to an expensive service, so to speak. Less demanding services could be used for transporting merchandise that must not be below 0 °C.

For defining new test for qualifying ATP units, old advice from reference books [7,8] on industrial experiments will be useful, as well as modern ways to reduce test uncertainties to commensurate levels [62,63].

Of course, each carriage could be optimised in a context of cost-benefit analysis. Note that the main cost comes from the engine and chassis (as cited before) but this element does not provide any specific support for the specialised transportation. The speciality is provided by the cooling machine acting inside the isolated body. The evaporator of the machine has to remove the heat load coming into the body through the walls. It is obvious that very powerful evaporators or heavily reinforced insulation will allow travel with an internal temperature inside the body much lower than that of open air; but those solutions are more expensive as either the evaporator or the insulator, or both, are made with improved performance. The cost-benefit analysis will help determine the features of those elements (cryo cycle and insulated body) in such a way that they can satisfy enough number of transportation asked for by clients. The owner of the carriage can accept an additional cost aimed at improving the thermal features, because the carriage would be able to attend a higher number of demanded services.

Such a detailed individual optimisation, which has a very positive effect in improving this whole commercial sector, is possible because the new proposed ATP is based on evaluating an engineering system by assessing the characteristic benefits of their units. In this way, ATP would really meet its goals.

## 8. Conclusions and Future Work

Transportation of perishable foodstuff and other products under controlled temperature is such an important activity that deserved 50 years ago to convene an international Treaty (which is managed through the Secretary-General office of the United Nations. The Treaty, however, is totally outdated in technology and is useless as a practical tool to rule that sector. Quality procedures and in-service conformity checks are also absent in the Treaty, what is a severe mismatch with the standards of modern engineering.

Fifty years ago, air conditioning was a luxury of the United States and North America, with very poor deployment in most of the world. Moreover, most of the cars made outside the USA did not include air conditioning.

Things have changed enormously, and now most of the new cars (and trucks, and buses, etc.) have air conditioning. At home, it is frequent to find a fridge for keeping meals and beverages slightly above 0 °C and a freezer reaching and keeping −20 °C. A so-called “cold chain” is now a social and commercial reality with only one weak link: transportation. Although some temperature recordings are mandatory in certain fields and certain countries, this is not included in the Treaty that still works on categories and classes, with a very poor representation of the physical reality.

It could be said that this commercial sector is a pending problem of optimisation in thermal engineering. To optimise is to get the most from an activity, using available resources in a reasonable way, at a commensurate cost. That was the subject of this article, where the most relevant items of thermal engineering in the ATP Treaty have been reviewed, to have a sound foundation to make a new proposal. It was found that we were enabled by different technologies to look for such optimum, and the costs of that leap forward seem very modest in comparison to the cost of transportation units in general and in comparison versus the benefits of keeping foodstuffs in good condition always. The word always was the root of proposing to select ATP bodies with active cooling machines. “Always” requires a continuous availability to produce a cryogenic effect of given specifications.

The next steps in this quest should be oriented to complete examples for having a systematic information on the type of services that would need controlled temperature transportation. Note that in some cases (meat, milk, and the like) the total amount transported in a year is enormous, but the unitary price is small. Therefore, the optimum solution is usually found in large trucks. On the contrary, pharmaceuticals and other small but more expensive pieces can be better managed in smaller trucks and vans.

Another field to be examined in this quest is the final distribution in cities and urban arrangements, where trucks cannot circulate. A model with 2 or 3 stages typically leads to decreasing sizes of the optimum carriage for each stage. The use of small containers organised from the beginning to the end of the distribution circuit, provides another degree of freedom that is in the very centre of optimising the result of this activity.

These suggestions for future work can seem obvious for many people, but the real fact is that the ATP sector is 50 years old, and it has not evolved in any direction. The technological updating proposed in this paper and a better qualification system of the thermal performance of the carriages could open a new business cycle and a more appealing field of engineering to work in.

## Figures and Tables

**Figure 1 entropy-23-00109-f001:**
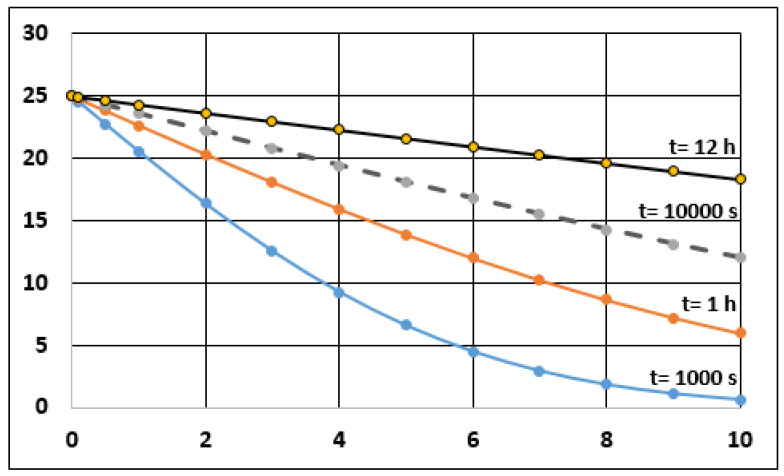
Evolution of the temperature profile inside the wall.

**Figure 2 entropy-23-00109-f002:**
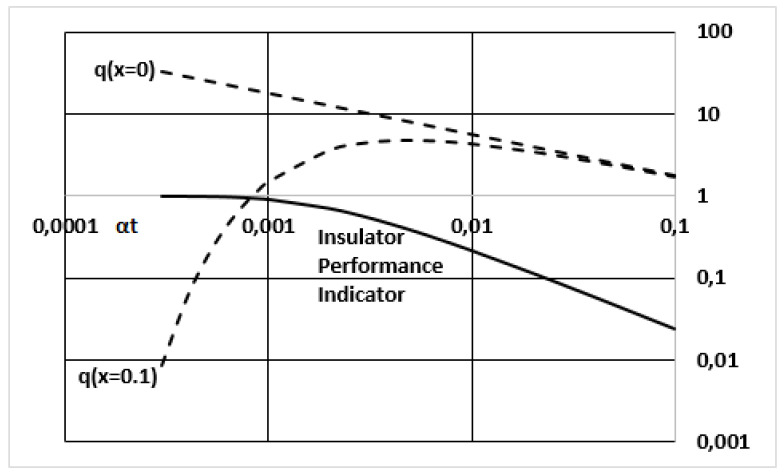
Evolution of the thermal fluxes in the outer face, *q*(*x* = 0),and the inner face of the panel, *q*(*x* = 0.1), and evolution along time of the Insulator Performance Indicator.

**Figure 3 entropy-23-00109-f003:**
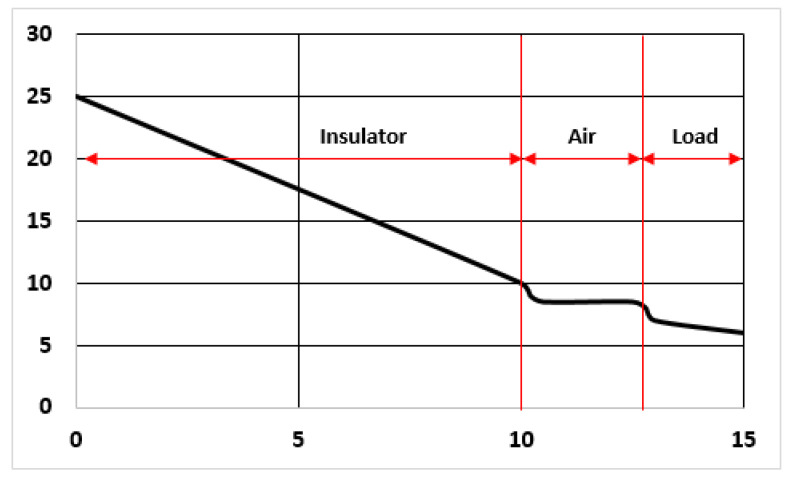
Temperature profile of a simply insulated container, with air filling the gaps of the chamber.

**Figure 4 entropy-23-00109-f004:**
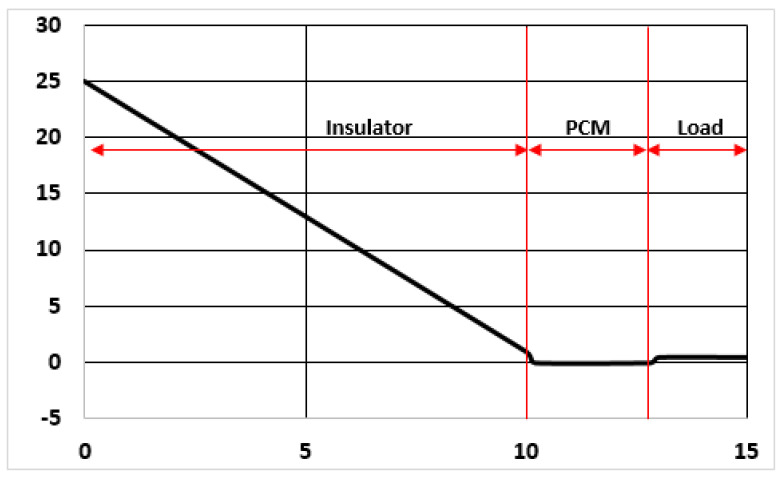
Temperature profile of an insulated container equipped additionally with Phase Change Material (ice) panels in the inner face of the outer insulator. In this case, the phase transition temperature is 0 °C. As long as the PCM remains frozen in part, the profile is kept so. Once finished an operational cycle, all PCM panels should be reloaded and/or cooled down until freeze.

**Figure 5 entropy-23-00109-f005:**
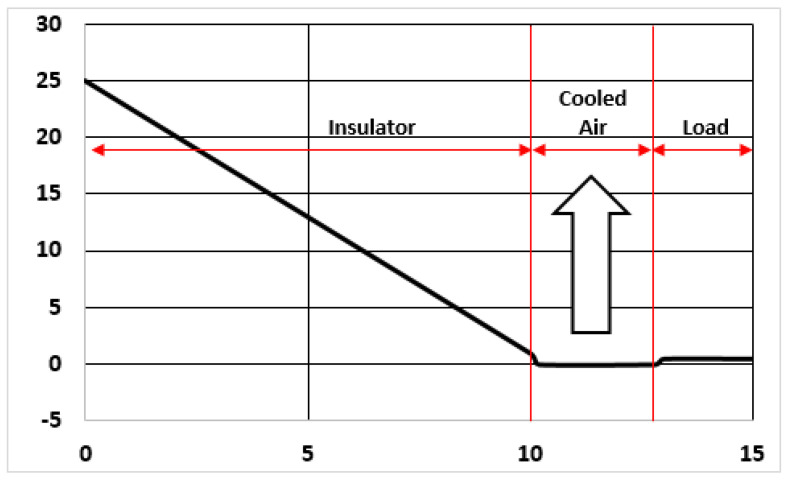
Temperature profile of a container equipped with active cooling, which primarily cools the air filling the gaps of the internal chamber. If necessary, active cooling can cool the load inside the container.

**Table 1 entropy-23-00109-t001:** Temperature evolution at a given point inside the wall, for different times.

*l*	*H*/*l*	1st Term	2nd Term	$T_{\left(H\right)}$−$T_{u}$/($T_{0}$−$T_{u}$)	*t* (s)
0.0010	100.0000	0.0000	0.0000	0.0000	1.75
0.0100	10.0000	0.0000	0.0000	0.0000	175.44
0.0200	5.0000	0.0004	0.0004	0.0000	701.75
0.0300	3.3333	0.0210	0.0184	0.0026	1578.95
**0.0350**	**2.8571**	**0.0513**	**0.0434**	**0.0080**	**2149.12**
0.0400	2.5000	0.0946	0.0771	0.0175	2807.02
**0.0500**	**2.0000**	**0.2076**	**0.1573**	**0.0503**	**4385.96**
0.0600	1.6667	0.3381	0.2386	0.0995	6315.79
0.0700	1.4286	0.4742	0.3124	0.1618	8596.49

**Table 2 entropy-23-00109-t002:** Result from tested vehicle YV2E4C4A0WB202032.

Ti	Te	DTi	DTe	PTV	PTR	PT	Dtie	Tm
32.0	8.4	1.1	0.8	102.0	475.0	577.0	23.6	20.2
32.0	8.4	1.1	0.8	103.0	472.5	575.6	23.6	20.2
100.0	100.0	100.0	100.0	101.0	99.5	99.8	100.0	100.0

**Table 3 entropy-23-00109-t003:** Result from tested vehicle VM3LVFS3F81R19409.

Ti	Te	DTi	DTe	PTV	PTR	PT	Dtie	Tm
32.0	8.5	0.5	0.9	939	1011	1949	23.5	20.25
32.0	8.6	0.5	0.9	940.4	999.9	1940.3	23.4	20.3
100.0	101.2	100.0	100.0	100.1	98.9	99.6	97.9	100.2

**Table 4 entropy-23-00109-t004:** Result from tested vehicle VS9S3G1408G147146.

Ti	Te	DTi	DTe	PTV	PTR	PT	Dtie	Tm
32.1	8.6	0.8	0.7	951.6	1206.6	2158.2	23.5	20.35
32.0	8.7	0.8	0.7	951.6	1193.7	2145.3	23.3	20.35
99.7	101.2	100.0	100.0	100.0	98.9	99.4	99.1	100.0

**Table 5 entropy-23-00109-t005:** Result from tested vehicle ZCFC358200D207782.

Ti	Te	DTi	DTe	PTV	PTR	PT	Dtie	Tm
32.0	8.5	1.0	0.4	107.6	242.9	350.5	23.5	20.25
32.0	8.5	1.1	0.4	107.4	247.2	354.6	23.5	20.25
100.0	100.0	110.0	100.0	99.8	101.8	101.2	100.0	100.0

**Table 6 entropy-23-00109-t006:** Result from tested vehicle WDB9044121P914406.

Ti	Te	DTi	DTe	PTV	PTR	PT	Dtie	Tm
32.0	8.5	0.7	0.4	106.9	251.5	358.4	23.5	20.25
32.0	8.5	0.7	0.4	107.2	252.6	359.8	23.5	20.25
100.0	100.0	100.0	100.0	100.3	100.4	100.4	100.0	100.0

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
