# Peer review of "Optimization of an Industrial Sector Regulated by an International Treaty. The Case for Transportation of Perishable Foodstuff"

_entropy, 2021, doi:10.3390/e23010109_

Round 1

Reviewer 1 Report

Authors have conducted deep literature analysis with the aim of presenting importance of observed theme and in the aim of providing quality recommendations for improvement of transportation of perishable foodstuff.

I would only recommend following:

  1. To move fullstop from title of subchapter 3.1 Retardants and Inuslators
  2. To provide other titles for tables 2-6. 
  3. In literature some sources don't have bolded year of publication.

Overall, the paper is well structured and it has been conducted an extensive analysis of ATP agreement and its lacks.

Author Response

All recommendations from the reviewers have been taken into account with the following exceptions:

1- We have not changed the structure of the paper, because it was specifically identified for presenting ATP features in a scientific community specialized in thermodynamics engineering, but not familiar with the ATP treaty. We consider that one of the aims of the manuscript is to draw attention on the importance of defining a set of optimized and updated technologies to improve the standards of an industrial sector that is lagging behind the evolution of thermodynamic engineering.

2- In fact, our answer to the 3rd Reviewer goes in the same direction: we think it is absolutely necessary a higher participation of this scientific community into the ATP sector. So, the manuscript we have presented here is aimed specifically to this community. ATP is well known in the circle of International Institute of Refrigeration, which has a technical subcomission called CERTE, which deals with ATP. But this technical community is not permeated from outside sufficiently in subjects of thermal engineering advancements.

The rest of the indications have been embodied in the new version, particularly a new abstract and new keywords.

Reviewer 2 Report

The whole article is written well, there are some minor changes that can improve the article.

The writing structure could change slightly with shorter explanations. all 7 sections should be categorized in 3 sections of introduction, methods, and results. avoid overexplaining. 

1- the abstract is not well organized to show what is done in your study. the style of abstract writing in the paper is like an introduction. it should prepare general information about your study, the procedure and at the end the general results.

2- please add two more keywords

3-for a paragraph between lines 44-52 a reference should be added.

Author Response

(The authors gave the same response as above.)

Reviewer 3 Report

Although the subject of this paper is very important and it is well written, this reviewer considers that Entropy is not the appropriate Journal to publish it. The International Journal of Refrigeration and the ASHRAE Journal are, for instance, more suitable to publish the paper.

Author Response

(The authors gave the same response as above.)
